



# Impact of surface and near-surface processes on ice crystal concentrations measured at mountain-top research stations

Alexander Beck[1], Jan Henneberger[1], Jacob P. Fugal[2], Robert O. David[1], Larissa Lacher[1], and Ulrike Lohmann[1]

[1]ETH Zurich, Institute for Atmospheric and Climate Science, Universitaetstrasse 16, 8092 Zurich, Switzerland
[2]Johannes Gutenberg-Universitaet Mainz, Institute for Atmospheric Physics, J.-J.-Becherweg 21, 55099 Mainz, Germany

*Correspondence to:* Alexander Beck (alexander.beck@env.ethz.ch) and Jan Henneberger (jan.henneberger@env.ethz.ch)

**Abstract.** In-situ cloud observations at mountain-top research stations regularly measure ice crystal number concentrations (ICNCs) orders of magnitudes higher than expected from measurements of ice nucleating particle (INP) concentrations. Thus, several studies suggest that mountain-top in-situ measurements are influenced by surface processes, e.g. blowing snow, hoar frost or riming on snow covered trees, rocks and the snow surface. A strong impact on the observed ICNCs on mountain-top

stations by surface processes may limit the relevance of such measurements and possibly affects the development of orographic clouds.

This study assesses the impact of surface processes on in-situ cloud observations at the Sonnblick Observatory in the Hohen Tauern Region, Austria. Vertical profiles of ICNCs above a snow covered surface were observed up to a height of $10\,\mathrm{m}$. The ICNC decreases at least by a factor of two at $10\,\mathrm{m}$, if the ICNC at the surface is larger than $100\,\mathrm{l}^{-1}$. This decrease can be up to

one order of magnitude during in-cloud conditions and reached its maximum of more than two orders of magnitudes when the station was not in cloud. For one case study, the ICNC for regular and irregular ice crystals showed a similar relative decrease with height, which cannot be explained by the above mentioned surface processes. Therefore, two near-surface processes are proposed to enrich ICNCs and explain these finding. Either sedimenting ice crystals are captured in a turbulent layer above the surface or the ICNC is enhanced in a convergence zone, because the cloud is forced over a mountain. These two processes would

also have an impact on ICNCs measured at mountain-top stations if the surrounding surface is not snow covered. Conclusively, this study strongly suggests that ICNCs measured at mountain-top stations are not representative for the properties of a cloud further away from the surface.



# 1 Introduction

Cloud microphysical properties (e.g. phase composition, cloud particle number concentrations and size distribution) next to dynamical processes are key parameters for the cloud's lifetime, the cloud extent and the intensity of precipitation they produce (Rotunno and Houze, 2007). In particular, orographic precipitation plays a crucial role for the world's water resources, as the headwaters of many rivers are located in alpine regions Roe (2015). In the mid-latitudes, mixed-phase clouds (MPCs) consisting of a mixture of ice crystals and supercooled liquid droplets, produce 30 to 50% of all land-falling (Mülmenstädt et al., 2015), because ice crystals quickly grow to precipitation size in the presence of supercooled liquid droplets. Thus, a good representation of orographic MPCs is crucial for accurate weather predictions in alpine terrain.

In-situ measurements are important to further improve our understanding of microphysical properties and fundamental processes of orographic MPCs (Baumgardner et al., 2011) and are frequently conducted at mountain-top research stations. Despite an improved understanding of the origin of ice crystals from nucleation (DeMott et al., 2010; Hoose and Möhler, 2012; Murray et al., 2012; Boose et al., 2016) as well as from secondary ice-multiplication processes (Field et al., 2017), the source of most of the ice crystals observed at mountain-top stations and their impact on the development of the cloud remains an enigma (Lohmann et al., 2016).

In-situ observations with aircraft usually observe ICNC on the order of $1\text{-}10\,\mathrm{l}^{-1}$ (Gultepe et al., 2001), whereas at mountain-top research stations (e.g. Elk Mountain, USA or Jungfraujoch, Switzerland) or near the snow surface in the Arctic ICNCs of several hundreds to thousands per liter are frequently reported (Rogers and Vali, 1987; Lachlan-Cope et al., 2001; Lloyd et al., 2015). Secondary ice multiplication processes like fragmentation (Rangno and Hobbs, 2001) or the Hallett-Mossop process (Hallet and Mossop, 1974) are usually ruled out as the source for the observed ice crystals due to the lack of large ice crystals necessary for fragmentation or a lack of large cloud droplets and the right temperature range necessary for the Hallet-Mossop processs. Instead, surface processes are proposed to produce such enormous ICNCs. Rogers and Vali (1987) suggested two possible processes as a source for the observed ICNC: riming on trees, rocks and the snow surface or the lifting of snow particles from the surface, i.e. blowing snow. In addition, Lloyd et al. (2015) suggested hoar frost as a wind independent, surface process to cause ICNCs larger than $100\,\mathrm{l}^{-1}$ for which they did not observe a wind speed dependency as expected for blowing snow. Although different studies are strife about the mechanisms to explain the measured high ICNCs, they agree on a strong influence by surface processes.

While the influence of surface processes on ICNC observed at mountain-top stations received more attention in recent years (Lloyd et al., 2015), the impact of surface processes on the development of supercooled orographic clouds, e.g. a more rapid glaciation and enhanced precipitation, has not been studied extensively (Geerts et al., 2015). If the proposed surface processes have the potential to impact the development of a cloud, depends primarily on the penetration depth of the reseuspended particle into a cloud, i.e. the maximum height above the surface to which the particles get lifted.

The height dependence of blowing snow has been studied in the context of snow redistribution ("snow drift") and reduced visibility due to the resuspended ice crystals by observing ice crystals up to several meters above a snow surface (Schmidt, 1982; Nishimura and Nemoto, 2005). It has been reported that blowing snow occurs above a certain wind speed threshold.





This threshold varies between $4\,\mathrm{m\,s^{-1}}$ and $13\,\mathrm{m\,s^{-1}}$ (Bromwich, 1988; Li and Pomeroy, 1997; Mahesh et al., 2003; Déry and Yau, 1999), because the concentration of blowing snow depends on snowpack properties (e.g. snow type, density, wetness,...) besides atmospheric conditions (e.g. wind speed, temperature, humidity,...) (Vionnet et al., 2013). Nishimura and Nemoto (2005) observed resuspended ice crystals from the surface up to a height of $9.6\,\mathrm{m}$ and found that the ICNCs usually decreased

to as low as 1-10 particles per liter. Meanwhile, during a precipitation event, the relative importance of the small ice crystals ($< 100\,\mu\mathrm{m}$) decreases from nearly $100\,\%$ at $1.1\,\mathrm{m}$ to below $20\,\%$ at $9.6\,\mathrm{m}$. The rapid decrease of ICNC with height observed in these studies may limit the impact of blowing snow on orographic clouds. The applicability of these results to orographic cloud may be restricted, because most of these studies were conducted in dry air conditions where ice crystals undergo rapid sublimation (Yang and Yau, 2008).

Lloyd et al. (2015) suggested that vapor grown ice crystals on the crystalline surface of the snow cover, i.e. hoar frost, may be detached by mechanical fracturing due to turbulence, independent of wind speed. To our knowledge only one modelling study exists, which assesses the impact of hoar frost on the development of a cloud. Farrington et al. (2015) implemented a flux of surface hoar crystals in the WRF (Weather Research and Forecasting) model based on a frost flower aerosol flux to simulate ICNCs measured at the Jungfraujoch by Lloyd et al. (2015). They concluded that the surface-based ice crystals have

a limited impact on orographic clouds because the ice crystals are not advected high into the atmosphere. However, more measurements of ice crystal fluxes from the snow covered surface are necessary to simulate the impact of surface-based ice crystal enhancement processes on the development of orographic clouds (Farrington et al., 2015).

In contrast to these findings, several remote sensing (i.e. satellite, lidar and radar) studies measured ice crystals advected as high as $1\,\mathrm{km}$ above the surface, which suggests an impact of surface-originated ice crystals on clouds. Satellite observations of

20 blowing snow from MODIS and CALIOP over Antarctica (Palm et al., 2011) observed layer thicknesses of ice crystals of up to $1\,\mathrm{km}$ with an average thickness of $120\,\mathrm{m}$ for all observed blowing snow events. Similar observations from lidar measurements exist from the South Pole with observed layer thicknesses of ice crystals of usually less than $400\,\mathrm{m}$, with some rare cases when a subvisual layer exceeded a height of $1\,\mathrm{km}$ (Mahesh et al., 2003). However, the suspension of clear-sky precipitation could not be ruled out as a source for the observed ice crystal layers. Observations of layers of ice crystals from radar measurements on

an aircraft in the vicinity of the Medicine Bow Mountains (Vali et al., 2012) observed ground-layer snow clouds which most of the time were not visually detectable but produced a radar signal. Re-suspended ice crystals from the surface or riming of cloud droplets at the surface can be the origin of the ice crystals in such ground-layer snow clouds. Geerts et al. (2015) presented evidence for ice crystals becoming lofted up to $250\,\mathrm{m}$ in the atmosphere by boundary layer separation behind terrain crests and by hydraulic jumps. They proposed that these ice crystals from the surface may lead to a rapid glaciation of supercooled

orographic clouds and enhanced precipitation. However, they also mentioned the limitation of radar and lidar measurements to separate the small ice crystals produced by surface processes from the larger falling snow particles and more abundant cloud droplets. They even concluded, that "to explore BIP (blowing snow ice particles) lofting into orographic clouds, ice particle imaging devices need to be installed on a tall tower, or on a very steep mountain like the Jungfraujoch".

In this study we assess the influence of surface processes on in-situ cloud observations at mountain-top stations and the

35 potential impact on orographic mixed-phase clouds. Vertical profiles of the ICNC up to a height of $10\,\mathrm{m}$ above the surface



were for the first time observed on a high-altitude mountain station with the holographic imager HOLIMO (Beck et al., 2017). HOLIMO is capable of imaging ice crystals larger than 25 µm and the shape of these ice crystals can be analyzed.

## 2  Field Measurements at the Sonnblick Observatory

### 2.1  Site description

This field campaign was conducted at the Sonnblick Observatory (SBO) situated at the summit of Mt. Sonnblick at 3106 m.a.s.l. (12°57′ E, 47°03′ N) in the Hohen Tauern National Park in the Austrian Alps. The SBO is a meteorological observatory operated all year by the ZAMG (Central Institute for Meteorology and Geodynamics). On the east and south the SBO is surrounded by large glacier fields with a moderate slope, whereas on the northeast a steep wall of approximately 800 m descends into the valley (Fig. 1, middle and right). Part of the SBO is a 15 m high tower used for meteorological measurements by the ZAMG.

The data presented in this paper was collected during a field campaign in February 2017.

### 2.2  Instrumentation

The properties of hydrometeors were observed with the holographic imager HOLIMO, which is part of the HoloGondel platform (Beck et al., 2017). The holographic imager was mounted on an elevator that was attached to the meteorological tower of the SBO (Fig. 2) to obtain vertical profiles of the hydrometeor properties up to a height of 10 m above the surface where the

platform was repeatedly positioned at five different heights as indicated in Figure 2. The holographic imager HOLIMO had a distance of approximately 1.5 m from the tower on the east-northeast side of the tower (Fig. 1, right).

The holographic imager HOLIMO yields single particle information, e.g. size, and shadowgraphs from all cloud particles in a three-dimensional volume on a single image, a so-called hologram. Per hologram, a sample volume of 16 cm³ with a length of 6 cm along the optical axis is examined. Concentrations and particle size distributions are calculated over 50 holograms

corresponding to a volume of 800 cm³. The open source software HoloSuite (Fugal, 2017) is used to reconstruct the in-focus images of the particles. Particles larger than 25 µm were classified as ice crystals based on the shape of their 2D image. For the calculaition of the ice water content (IWC), the circular euqivalent diameter is calculated based on the area of the ice crystals obtained from their 2D images, which is then used to calculate the mass of the ice crystals using a mass-diamater relationship given by Cotton et al. (2013). Similar to a study by Schlenczek et al. (2017) the ice crystals were further visually classified into

three different groups: regular, irregular and aggregates (Fig. 3). Because the visual classification of several thousands of ice crystals is time consuming this sub-classification of ice crystals was done only for the profiles obtained during the cloud-free time period on 4 February, 2017 and three profiles obtained on 17 February, 2017.

A single vertical profile was observed within a time interval of 10-12 min, by positioning the holographic imager HOLIMO at individual heights for 2 min and recording holograms at 4 frames per second. This results in a total volume of 8 liters of

sampled air for each vertical position.





Meteorological data are available from the measurements by the ZAMG, which include one minute averages of temperature, relative humidity and horizontal wind speed and direction at the top of the meteorological tower. Snow cover depth is manually observed by the operators of the SBO everyday. Based on these measurements, the change of the snow cover is calculated. This calculation includes all the changes of the snow cover depth, e.g. snow drift, sublimation, melting and fresh snow. Daily measurements of the total precipitation are available on the north and south side of the SBO. A ceilometer located in the valley north of the SBO measured the cloud base and cloud depth.

In addition, a 3D sonic anemometer was mounted at the top of the meteorological tower. However, data is only available occasionally, because most of the time the heating of the anemometer was insufficient in preventing the build up of rime on the measurement arms.

## 3  Results

The data presented was observed on 4 and 17 February, 2017. Figure 4 and 5 show an overview of the meteorological conditions on both days. The main difference is the wind direction, which was south-west on most of 4 February and north on 17 February.

### 3.1  Case study on 4 February, 2017

On 4 February a low pressure system moved eastwards from the Atlantic Ocean over northern France towards western Germany, where it slowly dissipated. Influenced by this low pressure system, the wind at the SBO predominantly came from the west-southwest with wind speeds between 10 and 25 $\mathrm{m\,s^{-1}}$ (Fig. 4). In the late afternoon, around 1900 UTC, when the low pressure system dissipated over western Germany, the wind direction changed to north and wind speeds decreased to a minimum of 5 $\mathrm{m\,s^{-1}}$. After 1900 UTC the wind speed increased again to up to 15 $\mathrm{m\,s^{-1}}$. Because no data is available from the 3D sonic anemometer, only one minute averages are available from the ZAMG measurements. The temperature did not change much during the day and stayed between -10 and -9 °C until 1900 UTC when the wind direction changed to north and the temperature decreased to -11 °C at 2200 UTC. The SBO was in cloud for most of the measurements, except for a short time interval between 1910 and 2020 UTC.

During all 24 profiles, the average ICNC reached a maximum of $200\,\mathrm{l^{-1}}$ at 2.5 m above the surface (Fig. 6) and decreased by a factor of two and four for the mean and median, respectively at a height of 10 m. This decrease in the ICNC with height suggests that surface processes strongly influence the ICNC close to the ground. For a more detailed presentation of the results, the measurement period is divided into four time intervals representing different meteorological conditions as indicated by the shaded areas in Figure 4. The most important features of the profiles for the different time intervals are summarized in Table 1.

Between 1910 and 2010 UTC (Fig. 7 third row), when the SBO was not in cloud, the ICNC reached its maximum of $600\,\mathrm{l^{-1}}$ at 2.5 m. The large shaded area represents the high variability of the ICNC over the measurement period. The average ICNC of all data decreases by a factor of 10 within 7.5 m and more than 98% of the observed ice crystals had irregular shapes (Fig. 8).

Between 2030 and 2200 UTC (Fig. 7, first row), when the SBO was in cloud again, the maximum ICNC is observed at a height of 4.1m and decreased on the same order of magnitude, by a factor of 9, up to a height of 10 m.





The highest ICNCs were observed in the time period between 1200 and 1500 UTC (Fig. 7 second row). The ICNC reached its maximum at 2.5 m with a mean value of $800 \, 1^{-1}$ and decreased by a factor of 2 within 7.5 m. Compared to the previous time interval the observed decrease is less, most likely due to the higher wind speeds of up to $25 \, \text{m s}^{-1}$, which lift the ice crystals higher up. Between different profiles the ICNC changed by a factor of up to 2, however, consistently for all profiles a decrease of ICNCs with height was observed.

In the morning between 8300 and 1100 UTC the observed ICNCs (Fig. 7, first row) are much lower than between 1200 and 1500 UTC, although wind speeds were as high as $20 \, \text{m s}^{-1}$. A possible reason is that the last snow fall was observed 3 days before the measurements. During this time, the loose ice crystals at the surface were blown away and the snowpack was solidified by temporal melting and re-freezing. Consequently, less ice crystals are expected to be re-suspended from the surface.

The correlation between wind speed and ICNC for 1-minute time intervals is shown in Figure 9. Instead of the average wind speed, the maximum wind speeds are used, because gusts, i.e. the highest wind speed in a time interval, are most relevant for re-suspending ice crystals from the surface. For the time interval between 0830 and 1500 UTC when the wind direction was west-southwest, no correlation is observed with wind speeds higher than $14 \, \text{m s}^{-1}$ (Fig 9, top). For the time interval between 1910 and 2230 UTC, when the wind direction was north, a much more pronounced dependency of the ICNC on wind speed is observed for wind speeds lower than $14 \, \text{m s}^{-1}$ (Fig 9, bottom row).

### 3.2 Case study on 17 February, 2017

On 17 February a cold front over northern Europe was moving southwards causing mainly northerly flow across the Alps and at the SBO (Fig. 5). Wind speeds observed at the SBO in the time interval between 1800 and 2000 UTC were between 5 and $10 \, \text{m s}^{-1}$. During this period the temperature decreased by 1 K from -12.5 to -13.5 °C. The SBO was in cloud starting at 1300 UTC with varying visibility between several meters up to several hundreds of meters. Some snow fall was observed in the afternoon between 1300 and 1500 UTC.

For this day, wind data from the 3D Sonic Anemometer were available, which allow a more detailed analysis of the correlation between the observed ICNC and wind speed. However, only four vertical profiles were obtained due to hardware problems with the computer. The first profile was measured in the morning at 1200 UTC when the SBO was not in cloud and no ice crystals were observed. Three more profiles were taken in the evening starting at 1800 UTC. For these profiles the ice crystals have been classified by hand and subclassified into three categories: regular, irregular and aggregates.

ICNCs of several hundreds per liter were observed in the lower height intervals and consistently decreased by a factor of 2 to 4 to a minimum below $150 \, 1^{-1}$ at 10 m (Fig. 10). The shape of the ice crystals are dominantly irregular, which is in good agreement with observations by Schlenczek et al. (2017). However, it is surprising that the fraction of irregular ice crystals stays constant with height (Fig. 12, right panels), because surface processes are expected to lift irregular ice crystals. With increasing height this impact is expected to decrease and the fraction of regular ice crystals, which originate from the cloud, are expected to increase. Thus, this observations is in contradiction to the expectations from surface processes. A further discussion of these results follows in section 4.1.





In contrast to the data of 4 February, no correlation between ICNC and the horizontal wind speed is observed (Fig. 11). This holds true for the 1-minute averages and maximum wind speeds observed by the SBO as well as the 1-second averages observed with the 3D sonic anemometer. However, ICNC increased with the vertical wind speed. The wind speed dependence of irregular and regular ice ICNCs is comparable, and both increase by a factor of 2 when the vertical wind speed increases

from 0-2 $\mathrm{m\,s^{-1}}$ to 4-6 $\mathrm{m\,s^{-1}}$ (Fig. 13). Whereas the shape of the size distribution of the irregular ice crystals hardly varies with height, larger regular ice crystals are more strongly reduced with height than smaller regular ice crystals (Fig. 14).

## 4 Discussion

### 4.1 Sources of enhanced ICNC observed at mountain-top research stations

To better understand the sources of enhanced ICNCs measured on 4 and 17 February, 2017, the results are compared to the
10 properties of blowing snow observed in the Arctic and in Antarctica.

#### 4.1.1 Impact of surface processes

For blowing snow two main layers are distinguished. In the saltation layer with a typical thickness of $0.01 - 0.02\,\mathrm{m}$ snow particles are lifted and follow ballistic trajectories. Depending on the particle size, the snow particles in the saltation layer either impact onto the snow surface or are transported by turbulent eddies into the suspension layer (e.g. Comola et al., 2017;
Gordon et al., 2009). The suspension layer height can be up to several 10s of meters, whereas the particle size above $1\,\mathrm{m}$ is usually smaller than $100\,\mathrm{\mu m}$ and particle concentration gradually decreases with height (Fig. 16 a). Two studies that observed the height dependence up to $10\,\mathrm{m}$ over a flat surface in the Arctic and in Antarctica are Nishimura and Nemoto (2005) and Mellor and Fellers (1986).

The measured ICNCs on 4 February, 2017 before 2030 UTC gradually decrease with height as expected from blowing snow
measurements in the Arctic and Antarctica. The wind from west-southwest during this time period (Fig. 4) transported the resuspended ice crystals from a area with a gentle slope towards the station. This explains why blowing snow similarly affects the height dependence of the ICNC close to the surface as observed over a flat terrain (Fig. 16 a). Mellor and Fellers (1986) reported the height dependence of the ice water content for different wind speeds up to a height of $10\,\mathrm{m}$. The observation between 1200 and 1500 UTC and between 1910 and 2020 UTC are compared to a parametrization by Mellor and Fellers
(1986) for the ice water content as a function of wind speed and height for blowing snow observed in the Arctic (Fig. 15). Both, the parametrization and the observation show a similar decrease with height, however, the measured water content is usually an order of magnitude less than expected from the parametrization. Between 1910 and 2020 UTC the SBO was not in cloud and the majority of the ice crystals had irregular habits (Fig. 8), consistent with the described surface process of blowing snow over a gentle slope.
A maximum of the ICNC at an elevated level as observed on 4 February, 2017 between 2030 and 2200 UTC or on 17 February, 2017, cannot be explained by the process of blowing snow over a gentle slope. It is proposed that elevated maximum



can be explained by the change in wind direction to north (Fig. 4). The terrain towards the north side of the station is steeper than to the west and the measurements were taken a few meter behind the mountain ridge (Fig. 1). Consequently a wind rotor is created which shadows the lower levels of the vertical profiles and possibly explains the elevated maximum of ICNC (Fig. 16 b), as observed on 4 February between 2030 and 2200 UTC (Fig. 7). The main contribution of blowing snow to the

5 observed ICNC is still expected to show up as irregular shaped particles and a decrease of the fraction of irregular ice crystals contributing to the total ICNC is expected with height above the given elevated level. This is in contrast to our observations on 17 February 2017, when the fraction of irregular to regular crystals remained constant with height.

### 4.1.2 Impact of near-surface processes

Two near surface-process are proposed to potentially modify the ICNC in the presence of a cloud near the surface. When a
10 cloud is moving over a mountain and cloud base is below the mountain top, a convergence zone of an enriched ICNC could develop when cloud particles below the mountain top are forced over the mountain (Fig. 16 d). If irregular and regular ice crystals are well mixed within the cloud below the mountain top, they will be consistently enriched in such a convergence zone. Thus, this near-surface process could explain the vertical profile of irregular and regular ICNC that was observed on 17 February, 2017.

Additionally, sedimenting ice crystals originating in cloud may be kept floating near the surface in a turbulent layer (Fig. 16 c), similar to the lifting of ice crystals in the suspension layer of blowing snow. However, in this case the sedimenting particles may have maintain their habits, because they have never reached the surface. Therefore, such an effect can enrich both, irregular and regular ice crystals near the surface, resulting in the same gradual decrease of the ICNC, while keeping the ratio of regular to irregular crystals constant.

The mechanisms illustrated in Figure 16 cannot only be considered separately, but can also occur as a combination with each other. For example, the observed decrease of the ICNC with height during the first two time periods on 4 February, 2017 (Fig. 7) is potentially a combination of blowing snow over a moderate slope (Fig. 16 a) with either a cloud or an additional enrichment by cloud ice crystals captured in a turbulent layer near the surface (Fig. 16 c). It is also important to note that the proposed near-surface processes (Fig. 16 c and d) can impact the observed ICNC at mountain-top research stations even
without a snow covered surface.

### 4.2 Wind dependence of the observed ICNCs

Turbulent eddies near the surface are responsible for the lifting of snow particles into the suspension layer (see sec. 4.1.1). Observations in the Arctic or Antarctica usually use wind measurements close to the surface ($< 3\,\mathrm{m}$) to estimate these turbulent eddies using the friction velocity. In this study only wind measurements on top of the meteorological tower at a height of 15 m
are available. Whereas for 4 February, 2017 only the horizontal wind speed averaged over 1-minute is available, on 17 February 1-second averages are available for horizontal and vertical wind speed.



A possible explanation for the observed dependence on vertical wind speed on 17 February (Fig. 13) is that the vertical wind speed determines the concentration of ice crystals lifted over a mountain ridge from the mountain-induced crystal convergence zone.

The dependence of the ICNC on horizontal wind speeds smaller than $14 \, \mathrm{m \, s^{-1}}$ on 4 February could be explained by a close relationship of the turbulent eddies and the observed horizontal wind speed. However, an explanation for the loss of this correlation for wind speeds larger than $14 \, \mathrm{m \, s^{-1}}$ is not possible with the available data. Potentially, the wind direction also has an influence on the observed wind dependence as it was west-southwest when no correlation was observed and north when a correlation was observed. Lloyd et al. (2015) also observed a dependence of the observed ICNC on horizontal wind speeds but only for a small fraction of cloud events (27% in 2013 and 13% in 2014). While they proposed blowing snow for the observation when a correlation was observed and hoar frost when this correlation was not observed, possible difficulties with using the horizontal wind speed as a proxy for the turbulence responsible for the resuspension of ice crystal from the surface could also curtain this expected relationship (Fig. 17):

Firstly, ice crystals are re-suspended from the surface by local turbulence and captured in the same air parcel (Fig. 17, right). For a certain time after their re-suspension the ice crystals are transported in the same air parcel as the local turbulence. However, with time they can leave this air parcel for example by sedimentation or turbulence. If the measurements are performed too far away from the place where the ice crystals were re-suspended, either the turbulence responsible for the re-suspension may already have dissipated or ice crystals already have been transported to other air parcels (Fig. 17, right). Both effects eliminate the correlation between the ICNC of re-suspended ice crystals and the responsible turbulence.

Secondly, the averaging time is crucial to observe the correlation between ICNCs and wind speed (Fig. 17, right). If the averaging time is too large, the correlation is averaged out. If the averaging time is too short, enriched ICNCs are also measured at lower wind speeds, because ice crystals have left their initial air parcel. In this study we chose an average time of 10-15 s, which is the expected time scale of gusts responsible for the re-suspension of ice crystals.

Thirdly, at mountain-top research stations local turbulence are also created by nearby structures. Since it is not possible to observe all of the turbulence, wind measurements at a height of 15 m above the surface are expected to be a good estimation of the strength of such turbulence.

Finally, the ICNC of re-suspended particles does not only depend on wind speed, but also on age of the snow cover and atmospheric conditions and a possible correlation may be suppressed in a data set with different snowpack and atmospheric conditions. Between 1900 and 2030 UTC on 4 February, 2017 when no cloud was present at the SBO a decrease of the ICNC was observed with time at a height of 2.5 m, possibly because the very loose ice crystals on top of the snow cover were blown away.

## 4.3 Origin of crystals measured at mountain-top stations

The origin of ice crystals observed at mountain-top research stations is an open question, because the ICNC exceeds the measured ice nucleating particle (INP) concentration by several orders of magnitudes (Fig. 18). Thus, additional processes,



i.e. ice multiplication (Field et al., 2017) as well as surface and near-surface processes, have to contribute significantly to the ICNC.

The contribution of ice crystals from the surface is on the order of several hundreds of ice crystals per liter, which is estimated from the measurements after 1910 UTC on 4 February, 2017. Without cloud (1910 – 2020 UTC), several hundreds of ice crystals of blowing snow were observed (see Sect. 4.1.1 and Fig. 16 a). With a cloud present (2030 – 2200 UTC), also several hundreds of ice crystals were observed near the surface and only several 10s to $100\,l^{-1}$ above 8.1 m. Assuming that the upper ICNCs are representative for the cloud, the contribution from the surface is similar.

An estimation of the impact of the proposed near-surface processes (Fig. 16) is difficult. The profiles observed on 17 February that are possibly affected by such a near-surface process still have a decreasing tendency in the upper levels. Therefore no information about the ICNC in cloud is available (Fig. 10). However, the contribution of near-surface processes is at least on the order of 100 - $250\,l^{-1}$. The ICNCs near the surface at the SBO (2.5 m) are comparable with similar measurements at the Jungfraujoch (JFJ) at a height of 2 m above the ground (Fig. 18) (Lloyd et al., 2015; Lohmann et al., 2016), which indicates a similar origin of the observed ice crystals at both stations.

The contribution of surface and near-surface processes of several hundreds per liter can explain most of the gap between the measured INP concentration and the observed ICNC (Fig. 18). Even at a height of 10 m the observed ICNCs of several 10s to $100\,l^{-1}$ exceed the expected INP concentration (Fig. 18). This discrepancy is either because even at a height of 10 m the cloud is influenced from the surface or ice multiplication processes contribute significantly.

### 4.4 Impact the surface on the ICNC in clouds

For the impact of surface and near-surface processes on the properties and the development of a cloud, the height dependence of the re-suspended ice crystal concentrations is crucial. If ice crystals are lifted only several meters of the surface, clouds are expected to be influenced only locally (Farrington et al., 2015). As most of profiles do not show a constant ICNC at the upper height levels, clouds are still affected by ice crystals from the surface. It is unclear if and how much clouds are influenced even at higher heights. For example, between 1200 and 1500 UTC (Fig. 7, second row), when wind speeds are higher than $20\,\mathrm{m\,s^{-1}}$ the ICNC at 10 m is still larger than $300\,l^{-1}$. Already a low concentration of ice crystals from the surface can have a significant impact on cloud properties, e.g. extent and lifetime. Therefore, a final statement of the impact on clouds is not possible.

### 5 Conclusions

This study assessed the impact of surface and near-surface processes on ICNC measured at mountain-top stations and possible implications on the atmospheric relevance of such measurements. For this, an elevator was attached to the meteorological tower of the SBO and vertical profiles of the ICNC were observed with the holographic imager HOLIMO on two days in February 2017. The main findings are:

- ICNC decreases with height. The ICNC near the ground is at least a factor of two larger than at a height of 10 m, if the ICNC near the ground is larger than $100\,l^{-1}$. The increase in ICNC near the ground can be up to an order of magnitude



during cloud events and even two orders of magnitudes during cloud free periods. Therefore, in-situ measurements of the ICNC at mountain-top research stations close to the surface overestimate the ICNC.

- Observations with a similar decrease of the concentration of irregular and regular ice crystals is observed with height, which cannot be explained by previously established surface processes, i.e. blowing snow. In the presence of a cloud, near-surface process are proposed that enriches the ICNC of irregular and regular ice crystals. Either sedimenting ice crystals are captured in turbulence near the surface or ice crystals are enriched in a convergence zone when a cloud is forced over a mountain. In both cases, the observed ICNC at mountain-top research stations is not representative for the cloud further away from the surface even without the presence of a snow covered surface.

- On 4 February, 2017 the observed ICNC shows a dependence on horizontal wind speed for wind speeds up to $14\,\mathrm{ms}^{-1}$. On 17 February a dependence of the ICNC on horizontal wind speed was not observed, but instead on vertical wind speeds. Possibly, horizontal or vertical wind speeds measured $15\,\mathrm{m}$ above the surface are not a good estimate for the turbulent eddies responsible for the re-suspension of blowing snow particles.

- The contribution of surface and near-surface processes to the observed ICNC at mountain-top research stations is estimated with several hundreds of ice crystals per liter. The ICNC in cloud without an contribution of surface and near-surface processes is estimated with several 10 s per liter, based on the observations between 2030 and 2200 UTC on 4 February 2017. This is still an orders of magnitudes higher than the expected INP concentration (Fig. 18). Additional processes, i.e. ice multiplication, has therefore also an significant contribution to the ICNC in clouds.

- The strong influence of surface and near-surface processes on the ICNC measured at mountain-top stations limits the atmospheric relevance of such mountain-top cloud measurements. However, the data set obtained is too small to make a clear statement under which conditions in-situ measurements at mountain-top research station may well represent the real properties of a cloud in contact with the surface and when not.

To better understand the processes that are responsible for enhanced ICNCs close to the surface and to further investigate the processes proposed in this study we suggest a more thorough field campaign with additional 3D sonic anemometers. They should be placed at the luv side of the ICNC measurement to measure the turbulence that may be responsible for the re-supsension of ice crystals, on the elevator and on the top of the tower. This may help to better understand the wind dependence of the ICNC and to find the origin of the observed ice crystals. At best three cloud imaging probes would be part of such a campaign and would be installed in parallel to the 3D sonic anemometers. In addition, to get a better estimate of the impact of re-suspended particle on cloud properties, especially for high wind speeds, the vertical profiles have to be extended to larger heights above the surface. Such a field campaign could be conducted using a tethered balloon system equipped with cloud imaging probes, which can be lifted several hundreds of meters into the atmosphere.

*Competing interests.* The authors declare that they have no conflict of interest.



*Acknowledgements.* The authors would like to thank Hannes, Olga, Fabiola and Monika for their assistance during the field campaign and Eric for his help with classifying ice particles. We also thank the head of the Sonnblick Observatory, Elke Ludewig, for access to the measurement site and the entire staff (Hermann Scheer, Norbert Daxbacher, Lug Rasser, Hias Daxbacher) for their hospitality and their excellent support and assistance during the field campaign. The meteorological measurements at the Sonnblick Observatory were provided

5   by the 'Zentralanstalt für Meteorologie und Geodynamik'(ZAMG). This project was supported by ETH Zurich under grant ETH-30 13-2.





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




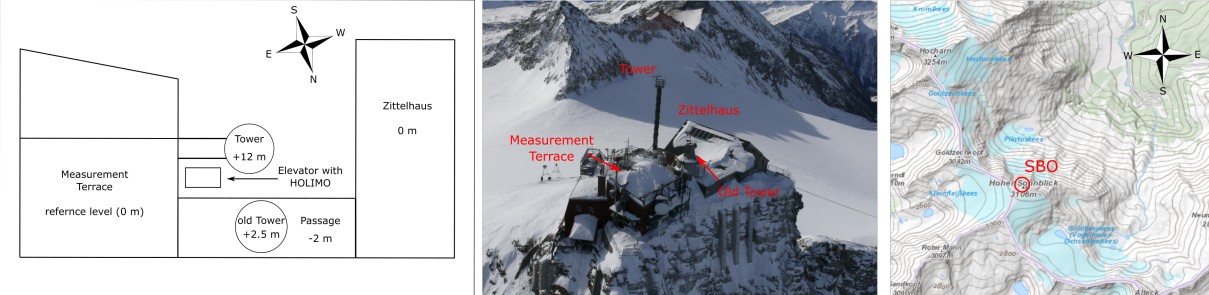

**Figure 1.** Sketch of the experimental setup and the surrounding structures (left) with their heights relative to the bottom of the measurement terrace. Aerial image of the Sonnblick Observatory (middle, courtesy by Michael Staudinger (ZAMG)) and a topographic map of the Hohen Tauern Region surrounding the Sonnblick Obersvatiory (right, basemap.at)

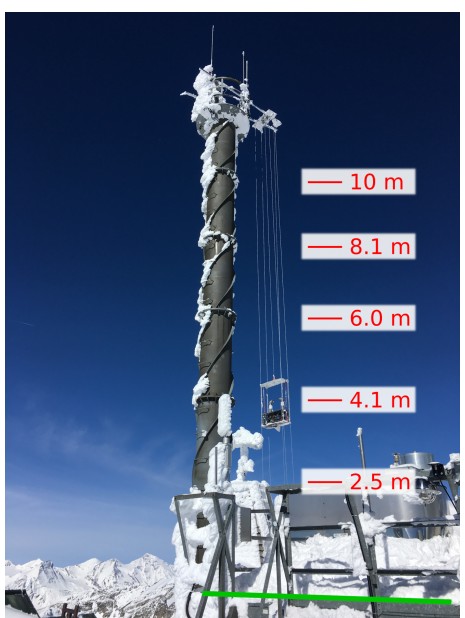

**Figure 2.** Set up of the elevator with the holographic imager HOLIMO mounted to the meteorological tower at the SBO (courtesy by Monika Burkert, ETH Zürich). The red lines and numbers indicate the five different heights where the elevator repeatedly was positioned to obtain vertical profiles of the ICNC. The reference height of $0.0\,\mathrm{m}$ is the bottom of the measurement platform (green line).





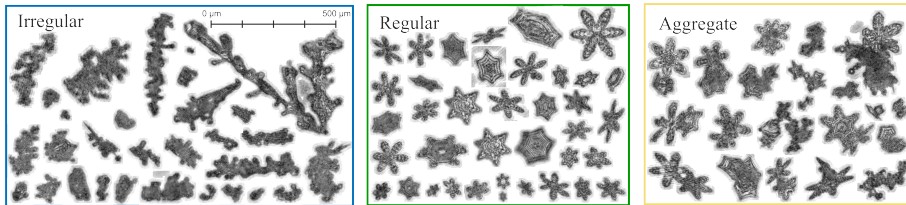

**Figure 3.** Examples of 2D images taken by the holographic imager HOLIMO for the three sub-classifications of ice crystal habits (irregular, regular and aggregates). The images are a collection recorded during the field campaign at different heights of the elevator.





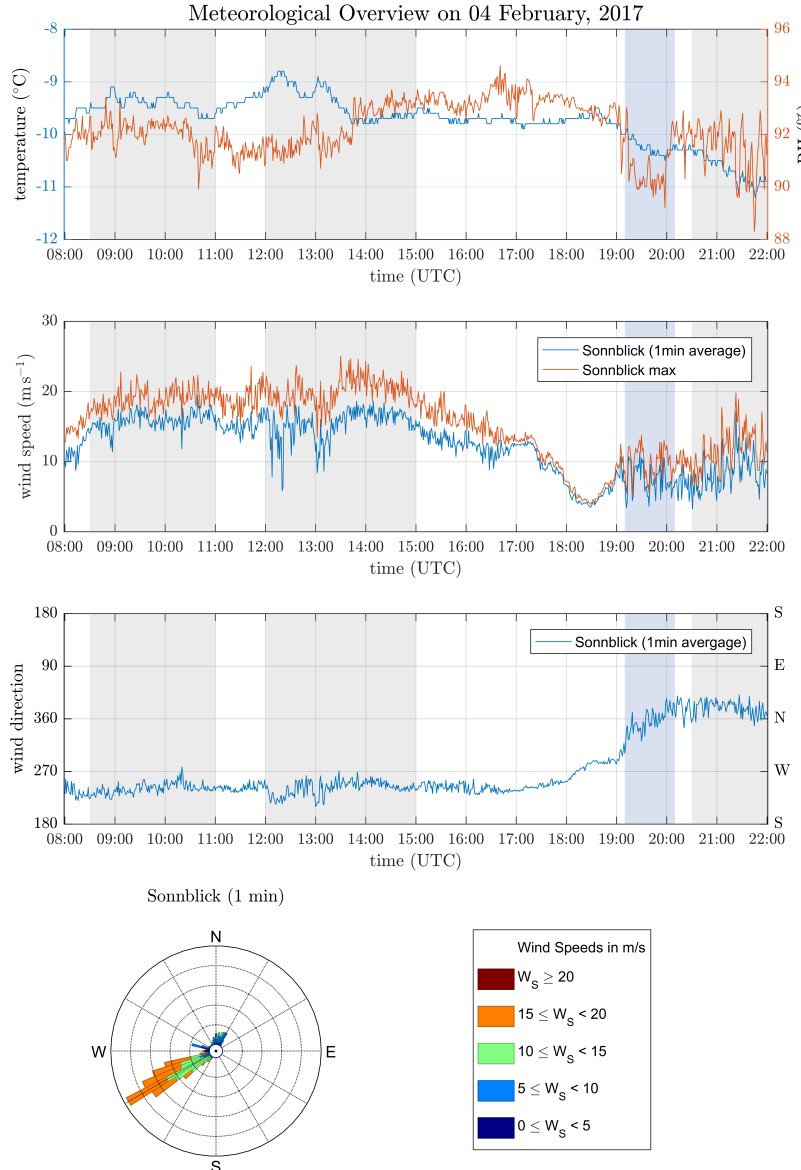

**Figure 4.** Overview of the meteorological conditions on 4 February, 2017 obtained from the SBO measurements. All measurements are 1-minute averages except for the maximum wind speed, which corresponds to the maximum wind speed observed during a 1-minute average. The shaded areas represent intervals with ice crystal measurements with the SBO in-cloud (gray), respectively not in-cloud (blue). Shown are the temperature and relative humidity (top), wind speed (second from top) and wind direction (second from bottom). A windrose plot is shown in the bottom panel.



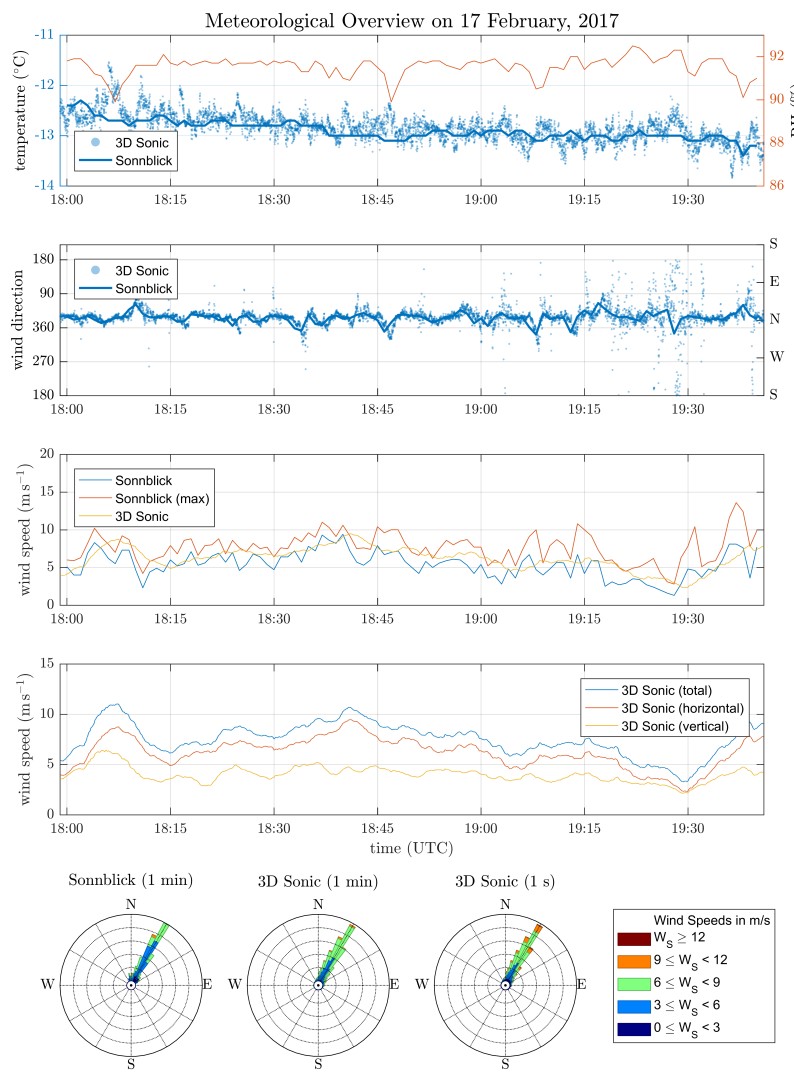

**Figure 5.** Overview on the meteorological conditions on 17 February, 2017 for the time interval when measurements exist (see Fig. 10). On this day temperature and wind measurements are available from the SBO and the 3D Sonic Anemometer. Shown are the temperature and relative humidity (top), wind direction (second from top), a comparison of the horizontal wind speed (middle) and detailed wind speed measurements from the 3D Sonic Anemometer (second from bottom). A windrose plot is shown in the bottom panel.




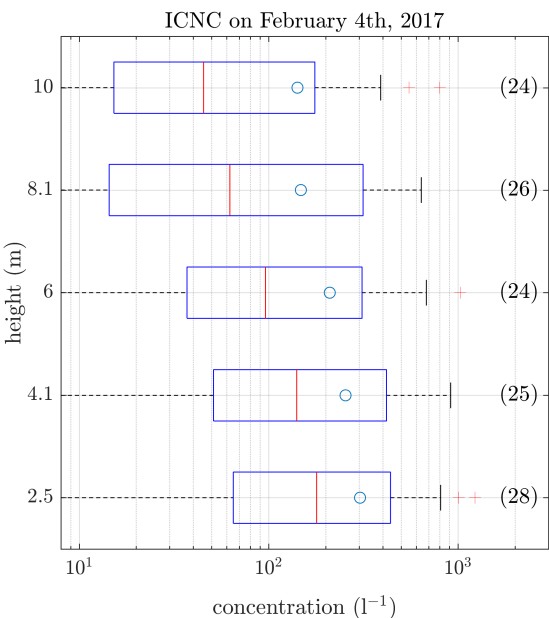

**Figure 6.** ICNC as a function of the height of the elevator at the meteorological tower of the SBO. This plot is a summary of the 24 profiles obtained on 4 February, 2017. The data was averaged for each height over the entire time period. The numbers in brackets are the amount of measurements per height. For each box, the central line marks the median value of the measurement and the left and right edges of the box represent 25th and the 75th percentiles, respectively. The whiskers extend to the minima and maxima of the data; outliers are marked as red pluses. The mean values of the measurements are indicated as blue circles.





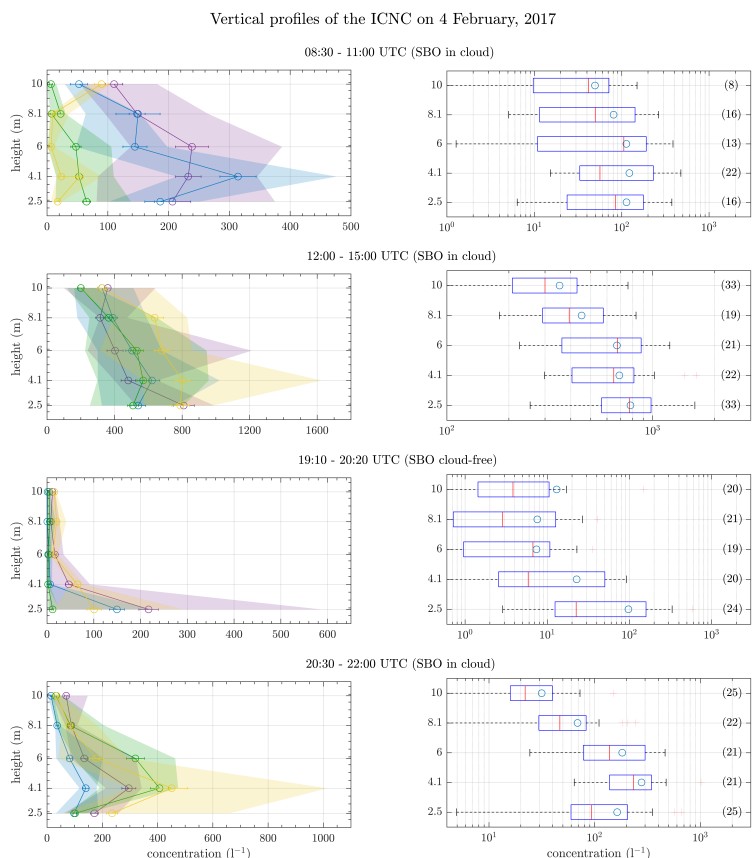

**Figure 7.** ICNCs as a function of height of the elevator for four different time intervals during 4 February, representing different conditions (Fig. 4). In the individual profiles (left), the circles indicate the mean and the error bars the standard error of the mean. The shaded areas extent from the minima to the maxima of the measured ICNC. The box plots (right) show a summary of all profiles in the respective time interval as in Figure 6.





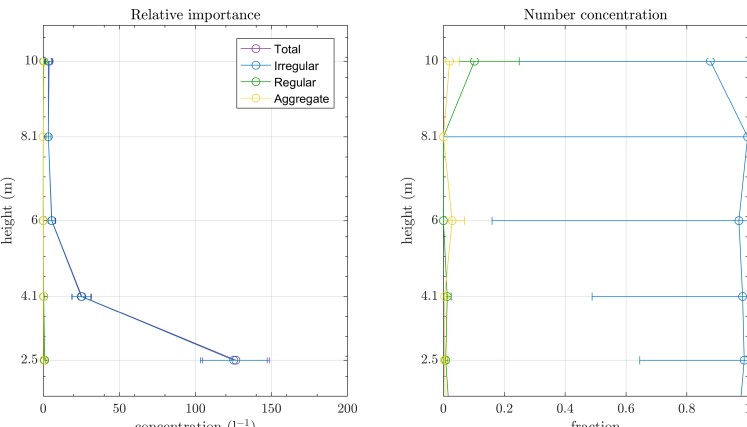

**Figure 8.** Vertical profile of the concentration (left) and the fraction (right) of individual ice crystal habits for the profiles between 19:10 and 20:20 UTC on 04 February, 2017. The concentration of regular crystals and aggregates is below $1\,l^{-1}$ for all heights. For the fraction, the ICNC of individual habits were divided by the total ICNC. The circles represent the mean and the error bars represent the standard error of the mean.




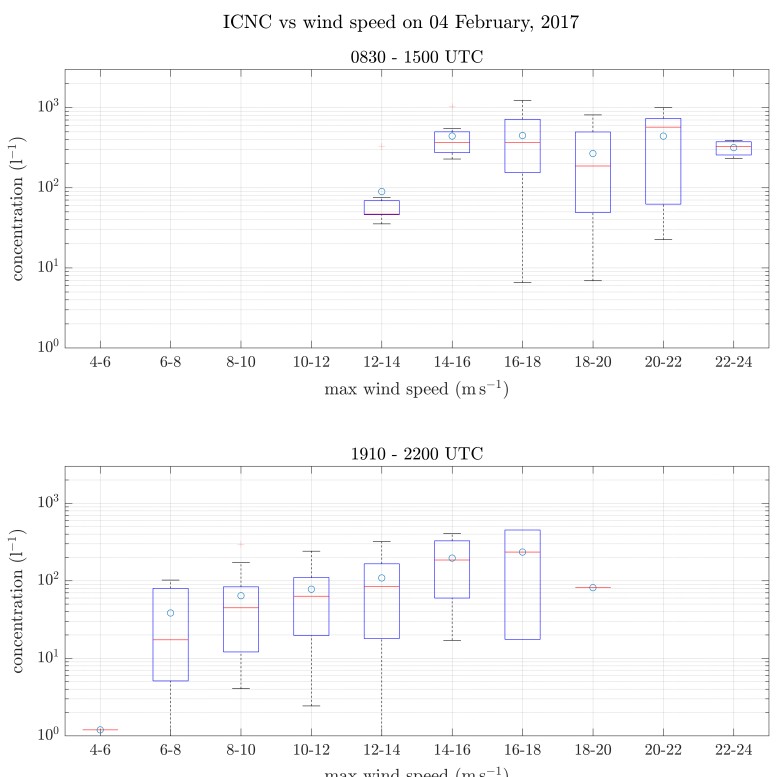

**Figure 9.** As Figure 6, but for ICNC as a function of the horizontal wind speed for the time periods between 0830 and 1500 UTC when the wind direction was from the west-southwest (top) and between 1910 and 2200 UTC when the wind direction was from the north (bottom). The ICNCs from HOLIMO are 1-minute averages and the wind speeds from the SBO are the maxima in the respective one minute intervals.

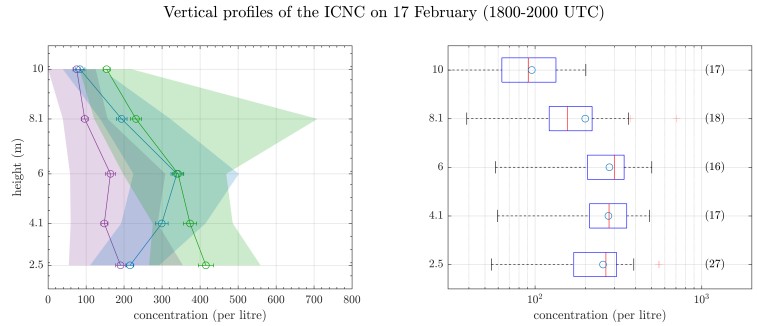

**Figure 10.** As Figure 7, but for ICNC as a function of height observed on 17 February, 2017.





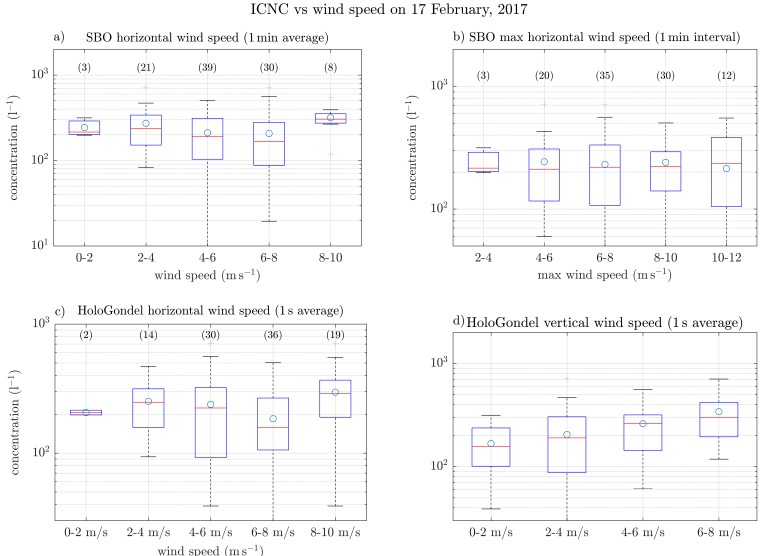

**Figure 11.** As Figure 9 only for 17 February and four different wind speed measurements: a) 1-minute averages of the horizontal wind speed from the SBO, b) maximum wind speed of the corresponding time interval in a), c) 1-second averages of the horizontal wind speed and d) 1-second average of the vertical wind speed both from the 3D Sonic Anemometer.

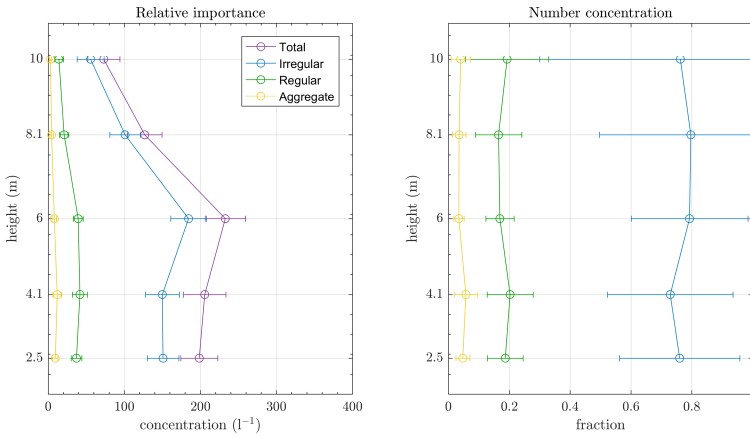

**Figure 12.** Vertical profile of the concentration (left) and the fraction (right) of individual ice crystal habits for the profiles on 17 February, 2017. For the fraction, the ICNC of individual habits were divided by the total ICNC. The circles represent the mean and the error bars represent the standard error of the mean.





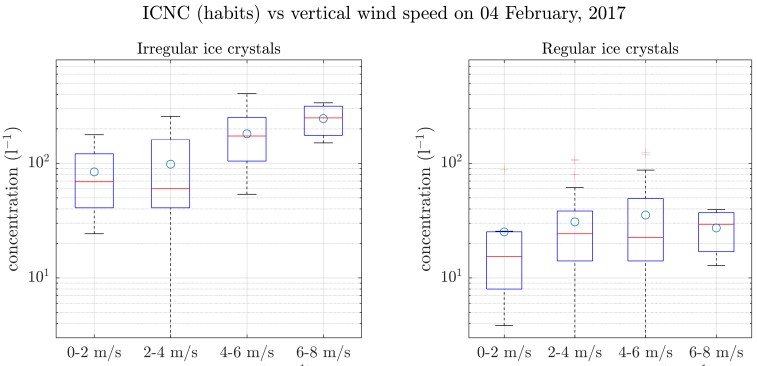

**Figure 13.** As Figure 9 only for 17 February, 2017, for the ICNC of different ice crystal habits as a function of the vertical wind speed. Aggregates are not shown because of their very low concentrations.

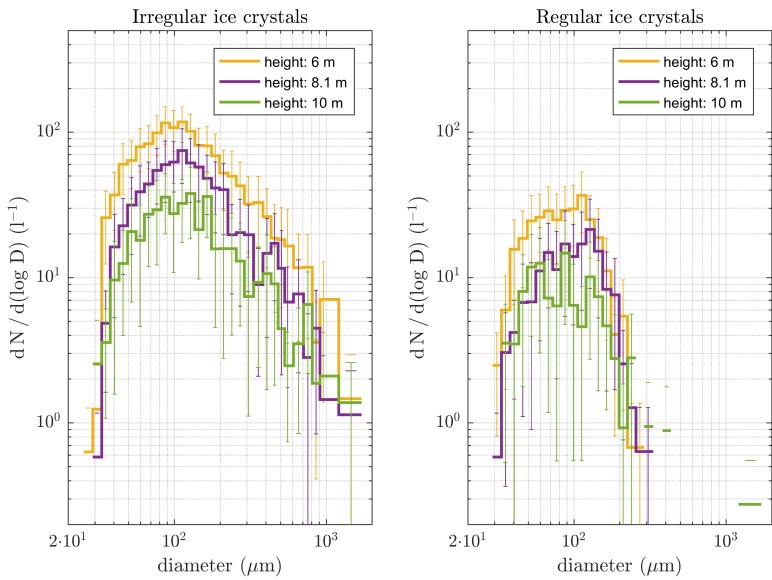

**Figure 14.** Number size distribution of the irregular (left) and regular (right) ice crystals observed on 17 February, 2017 as a function of height. The error bars represent the standard error of the mean. Aggregates are not shown because of their low concentrations.



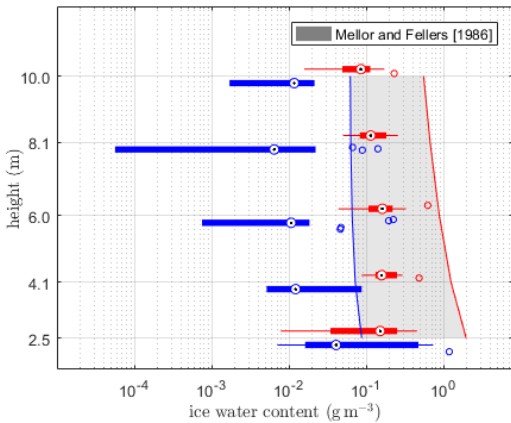

**Figure 15.** Comparison of the height dependence of the ice water content for the time intervals between 1200 and 1500 UTC (red boxplots) and 1910 and 2020 UTC (blue boxplots) using the blowing snow parameterisation of Mellor and Fellers (1986). The wind speed in the time interval between 1200 and 1500 UTC was around $20\,\mathrm{m\,s^{-1}}$ and between 1910 and 2020 around $10\,\mathrm{m\,s^{-1}}$. The grey shaded area indicates the calculated ice water content from the parameterisation for wind speeds between 10 (blue line) and $20\,\mathrm{m\,s^{-1}}$ (red line).

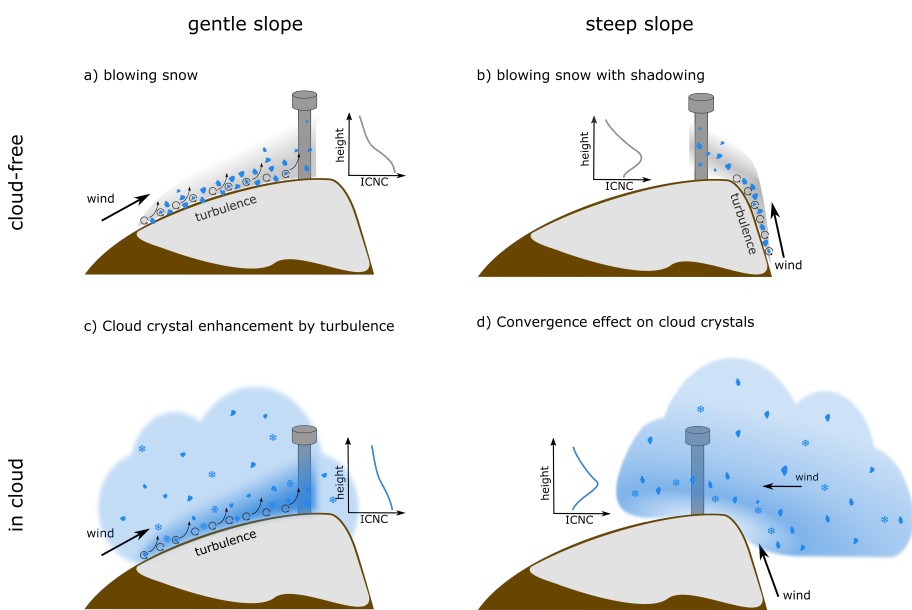

**Figure 16.** Illustration of surface and near-surface processes, which impact the measured ICNC at mountain-top research stations. a) and b) illustrate the difference in the height dependence of blowing snow over a surface with a gentle and a a convergence effect on the ICNC of regular and irregular ice crystals if a cloud is forced over a mountain.



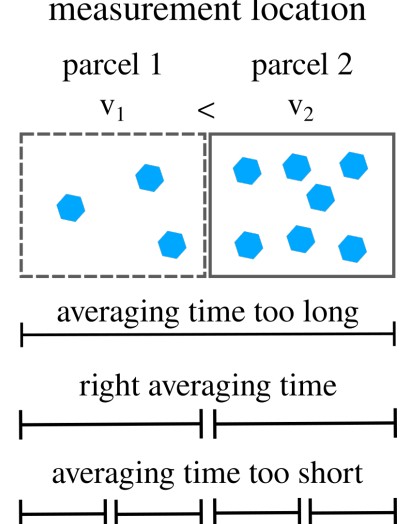

**Figure 17.** Illustration of the challenges observing the wind dependence of blowing snow (Sec. 4.2). The two squares represent two air parcels with a duration $\Delta t$ of $10 - 15\,\mathrm{s}$, which is typical for a gust. Parcel 2 represent a gust with a higher windspeed $v_2$ than the average windspeed $v_1$ in parcel 1. At the location of particle lifting (left) ice crystals are lifted due to the high turbulence in air parcel 2. On the way to the measurement location (right) some of the ice crystals were transported to the other air parcel, e.g. by sedimentation or turbulence. Also the effect of different averaging times is illustrated (right).





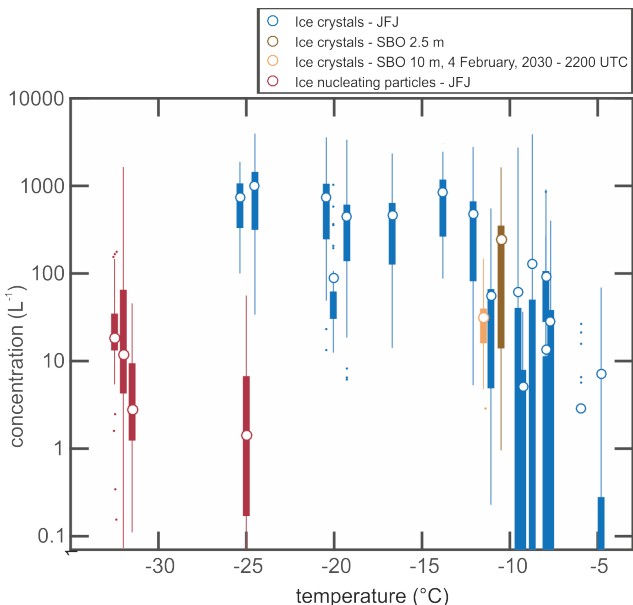

**Figure 18.** Ice crystal number concentrations (ICNCs, orange and brown) measured on the elevator at the SBO are compared to the ICNCs (blue) and ice nucleating particle (INP) concentrations (red) measured at the High Altitude Research Station, Jungfraujoch (JFJ). The ICNCs at the JFJ were measured with HOLIMO 2 during the winter 2012 and 2013 (Lohmann et al., 2016). Each box represent a cloud case. The INP concentrations were measured with the Horizontal Ice Nucleation Chamber (Lacher et al., 2017, and additional measurements) at relative humidity with respect to water ($RH_w$) of $103 - 104$ %. The three boxes on the left were taken during (from the left) Sahara dust events, summer seasons and winter seasons at a temperature of -31 °C, but shifted slightly to visualize them. The measurement at -25 °C were taken during a summer season. The left and right edges of each box represent 25th and the 75th percentiles, the circle the mean value and the small dots are outlier.





**Table 1.** Summary of important features of the ICNC profiles observed on 4 February (Fig. 7) and 17 February (Fig. 10), 2017. $\overline{ICNC}_{max}$ refers to the observed maximum of the mean ICNC over height. $\overline{ICNC}_{10}$ refers to the average ICNC at 10 m.

| time interval (UTC) | height of $\overline{ICNC}_{max}$ (m) | $\overline{ICNC}_{max}$ ($l^{-1}$) | $\overline{ICNC}_{10}$ ($l^{-1}$) | processes involved from Fig. 16 |
|---|---|---|---|---|
| 4 February, 2017 | | | | |
| 0830 - 1100 | 4.1 | 150 | 50 | a),c) |
| 1200 - 1500 | 2.5 | 800 | 250 | a),c) |
| 1910 - 2020 | 2.5 | 100 | 10 | a) |
| 2030 - 2200 | 4.1 | 300 | 30 | b),d) |
| 17 February, 2017 | | | | |
| 1800 - 2000 | 4.1-6 | 300 | 100 | b),d) |