# Peer review of "Impact of surface and near-surface processes on ice crystal concentrations measured at mountain-top research stations"

_Atmospheric Chemistry and Physics, 2017_

## Referee Comment (RC1) · Anonymous Referee #2 · 17 Dec 2017

Paper is well written and scientifically good. I recommend to publish the paper in current version.

---

## Referee Comment (RC2) · Anonymous Referee #1 · 29 Jan 2018

**Review of 'Impact of surface and near-surface processes on ice crystal concentrations measured at mountain-top research stations'**

The authors present observations of near-surface ice crystal concentrations from the Sonnblick Observatory, Austria. The results are important in the context of recent measurements at mountain top sites that suggest high ice crystal (100s or 1000s L$^{-1}$) that cannot be explained by existing primary ice nucleation schemes and that are also hard to attribute to secondary ice production processes given our current understanding.

Although the dataset is limited I think the observations are important and worth publication given the revisions suggested.

**Specific Comments**

I think the microphysical processes relevant to the work presented should be more clearly described or expanded on in the introduction. Relevant references are included, but I think it would be good to briefly describe in a bit more detail some of the main secondary ice processes in free floating cloud e.g. mechanical break up, rime-splintering, drop shattering. This helps the reader understand some of the mechanisms that are already thought to enhance ice concentrations.

I'd like to see if it's possible to look at only pristine ice crystals or only irregular ice crystals vs wind speed are the dependencies different? I think you say that in general the ratio of irregular to regular ice crystals stays similar – but I don't think this is the case looking at the habit segregated figures of ICNC concentration vs altitude.

P6 L30 – This paragraph only really holds true with some pretty big assumptions, no irregular ice crystals are produced in cloud and are therefore only produced from the surface, and that pristine ice crystals are all produced in cloud with no contribution from the surface. Although still very early research I believe there's increasing evidence for pristine ice crystals generated from the surface – though the exact physical mechanisms and the optimum conditions for this to take place is still unclear. The paragraph is also confused by the previous statement that the SBO is out of cloud.

Are there any useful references to convergence zones as described in section 4.1.2? I think convergence zones and sedimenting ice crystal theories need a much more thorough discussion, possibly under their own sub section headings. In its current form I don't find the explanations very well backed up.

ICNCs could be added to the microphysical time series figures.

There's 2 different wind measurements – If possible I'd like to see a comparison between the two where available.

Were the clouds glaciated/mixed phase at the site? Is there any information on the liquid phase from the holography?

What were the reasons for the dataset being limited to 2 events?

English is generally good, but the manuscript should be carefully checked as there were several grammatical/spelling mistakes.

**Technical Corrections/Further Comments**

P1 L3-4 These all refer to secondary ice processes? It's worth stating this.

P1 L5 relevance with respect to which processes? Primary ice nucleation? Secondary ice processes? I think that you are correct - the measurements at these sites are definitely complicated by the potential for surface generated ice particles.

P1 L15 Agreed - they are not representative when compared with free floating clouds away from ice surfaces, but it is important to consider potential impacts of surface ice processes on clouds above these surface, whether in contact or close enough to be influenced.

P2 L2 distribution(s)

P2 L6 Precipitation?

P2 L7 The bergeron findiesen process should be stated here.

P2 L15 primary ice concentrations

P2 L20 'lack of large'

P3 L14 I think the other important conclusions from Farrington et al (2015) could be described here including the finding that secondary ice could not account for the concentrations in the model

P3 L23 subvisible

P5 L12 'northerly'

P8 L17 'Maintained their habits, because they don't reach the surface'

P8 L25 Has this been studied over ice/snow free surfaces?

P9 L12 – is curtain supposed to be curtail?

P10 L20 of should be 'off' the surface.

P11 L16-17 Sentence needs rephrasing

P11 L24 poor sentence with grammatical/spelling mistakes

P11 L28 particle should be 'particles'

Figure 7 – what are the different colours for shading? I assume it's regular, irregular and aggregates, but what is purple?

---

## Author Comment (AC1) · 13 May 2018

We would like to thank Anonymous Referee #1 for carefully reading our manuscript and suggesting to publish the manuscript in the current version.
* * *

---

## Author Comment (AC2) · 13 May 2018

We would like to thank the Anonymous Referee #1 for having reviewed this paper and his valuable comments and suggestions. We answer each of them hereafter (bold black) and add when needed the modifications in the revised version of the manuscript (bold blue).

**I.) Point-by-point response to specific comments**

1) I think the microphysical processes relevant to the work presented should be more clearly described or expanded on in the introduction. Relevant references are included, but I think it would be good to briefly describe in a bit more detail some of the main secondary ice processes in free floating cloud e.g. mechanical break up, rime-splintering, drop shattering. This helps the reader understand some of the mechanisms that are already thought to enhance ice concentrations.

**We added a brief description of secondary ice mechanisms in free floating clouds and the production of ice crystals from surface processes to the introduction.**

**P2 L21:** This discrepancy between ice nuclei and ICNC may be explained by so-called secondary icemultiplication processes. A commonly accepted secondary ice-multiplication process to enhance ICNCs in free floating clouds is the rime-splintering or Hallett-Mossop process. This process describes the production of small splinters after the impact of cloud droplets on ice crystals and a subsequent burst of the cloud droplet during its freezing process. It is active only in a small temperature range between -3 to -8 \_C and the presence of small (<~ 13 μm ) and large (>~ 25 μm ) cloud droplets is required (Hallett and Mossop, 1974; Choularton et al., 1980). Another secondary ice-multiplication process is the fracturing of fragile ice crystals upon collision with other solid cloud particles (Vardiman, 1978; Griggs and Choularton, 1986). Although this process has been studied in the lab and is expected to occur at temperatures of ~ -15 °C , there is little evidence from field measurements for this process to significantly contribute to the ICNC (e.g. Lloyd et al., 2014; Crosier et al., 2011; Crawford et al., 2012). Other processes that produce secondary ice crystals are associated with the freezing of cloud droplets and subsequent break-up or the ejection of small spicules (Lauber et al., 2018).

**P3 L3:** Riming as a surface process is similar to the previously described rime-splintering process in free floating clouds. For this process to be active, cloud droplets need to be present near the surface, as typically the case with orographic mixed-phase clouds.

**P3 L8:** Hoar frost describes the formation of vapor grown ice crystals on the crystalline snow surface, which may be detached due to mechanical fracture.

2) I'd like to see if it's possible to look at only pristine ice crystals or only irregular ice crystals vs wind speed are the dependencies different? I think you say that in general the ratio of irregular to regular

ice crystals stays similar – but I don't think this is the case looking at the habit segregated figures of ICNC concentration vs altitude.

Figure 13 in the submitted manuscript shows the wind speed dependence of only pristine ice crystals and only irregular ice crystals of the measurements on 17 February 2017. As described in section 3.2 both show an increase of the ICNC by approximately a factor of 2 if the vertical wind speed increases from 0-2ms-1 to 4-6ms-1. Figure 12 of the submitted manuscript also shows that the ratio of irregular to regular ice crystals stays similar for the different levels of the elevator. While the pristine ice crystals contribute with 20% to the total number the irregular ice crystals contribute with approximately 80%.

3) P6 L30 – This paragraph only really holds true with some pretty big assumptions, no irregular ice crystals are produced in cloud and are therefore only produced from the surface, and that pristine ice crystals are all produced in cloud with no contribution from the surface. Although still very early research I believe there's increasing evidence for pristine ice crystals generated from the surface – though the exact physical mechanisms and the optimum conditions for this to take place is still unclear. The paragraph is also confused by the previous statement that the SBO is out of cloud.

It is true, that these assumptions are questionable. Therefore, we added a paragraph to the discussion section to back up our thoughts on these assumptions. We also added the statement, that a significant contribution of regular ice crystals produced from surface processes can't be excluded and vice versa irregular crystals also originate in cloud. However, the separation in irregular and regular shaped ice crystals is realized as an additional analysis of possible mechanisms to enhance ICNCs near the surface.

**P1 L10:** For one case study, the ICNC for regular and irregular ice crystals showed a similar relative decrease with height. This suggests that either surface processes produce both irregular and regular ice crystals or other effects modify the ICNCs near the surface.

**P7 L28:** To disentangle possible sources and mechanisms, which enhance the observed ICNCs at mountain-top research stations, the following discussion will be based on the observed height profile of the ICNC and the observed ice crystal shape.

In the context of snow redistribution blowing snow has been studied thoroughly. For blowing snow, two main layers are distinguished. In the saltation layer, with a typical thickness of 0.01 - 0.02 m, snow particles are lofted and follow ballistic trajectories. Depending on the crystal size, the crystals in the saltation layer either impact on to the snow surface or are transported by turbulent eddies into the suspension layer (e.g. Comola et al., 2017; Gordon et al., 2009), which can extend up to a height of several 10s of meters above the surface. Nishimura and Nemoto (2005) and Mellor and Fellers (1986) observed the height dependence of blowing sow up to 10m over a flat surface in the Arctic and in Antarctica and found that particles reaching layers higher than 1 m above the surface are usually smaller than 100  $\mu$ m and the particle concentration gradually decreases with height (Fig. 16 a). Similar to blowing snow we expect such a height dependence for any other surface process. As such, a gradual decrease of ICNCs with height is expected for any surface process and no height dependence is expected for ice crystals produced in free floating clouds.

While ice crystals observed in free floating clouds have mainly (> 80%) irregular habits (e.g. Korolev et al., 1999, 2006; Wolf et al., 2018), no studies have investigated the ice crystal shape produced by surface process like hoar frost, blowing snow or riming on trees, rocks or the snow surface. We expect irregular shapes for re-suspended ice crystals, i.e. blowing snow, due to mechanical fracturing upon their impact on the surface or due to successive melting and freezing of the ice crystals on the snow surface.

Ice crystals originating as hoar frost grow in regular shapes on the snow surface. If these vapor grown ice crystals keep their regular shape depends on the exact physical process how they are detached from the surface. While some ice crystals may keep their initial regular habit, for other ice crystals this regular habit may be destroyed when they are detached from the surface due to mechanical fracturing as described by Lloyd et al. (2015). Similar to blowing snow, the ICNC from hoar frost is likely to be increased near the surface, because only smaller ice crystals are lofted higher up. In this layer ice crystals are likely to collide and fracture. On the one hand, this reduces the probability to observe regular ice crystals from surface processes. On the other hand, if small regular and irregular ice crystals ( $^{\mu}\mu$ ) are produced, they have the potential to grow into larger regular shaped ice crystals being observed at the measurement location.

4) Are there any useful references to convergence zones as described in section 4.1.2? I think convergence zones and sedimenting ice crystal theories need a much more thorough discussion, possibly under their own sub section headings. In its current form I don't find the explanations very well backed up.

The ideas of a convergence zone and sedimenting ice crystals to describe the observed profiles of regular and irregular ice crystals is new to our knowledge. Therefore, we can't provide any references to back them up. We see these explanations only as an alternative to possible influences from the surface and don't want to state that these ideas are the final explanations. We see section 4.1.2 more as a stimulation for further investigation of these idea.

5) ICNCs could be added to the microphysical time series figures.

We included the microphysical time series to the Figures 4 and 5.